# Polymer-Assisted Synthesis, Structure and Magnetic Properties of Bimetallic FeCo- and FeNi/N-Doped Carbon Nanocomposites

Gulsara D. Kugabaeva [1,2], Kamila A. Kydralieva [1,*], Lyubov S. Bondarenko [1], Rose K. Baimuratova [2],
Dmitry Yu. Karpenkov [3,4], Ekaterina A. Golovkova [5], Pavel N. Degtyarenko [5], Nina D. Golubeva [2],
Igor E. Uflyand [6] and Gulzhian I. Dzhardimalieva [1,2,*]

[1] Department of Materials Science, Moscow Aviation Institute (National Research University), Volokolamskoe Shosse, 4, Moscow 125993, Russia; gulsara_kugabaev@mail.ru (G.D.K.); l.s.bondarenko92@gmail.com (L.S.B.)
[2] Federal Research Center of Problems of Chemical Physics and Medicinal Chemistry, Russian Academy of Sciences, Semenov Avenue, 1, Moscow Region, Chernogolovka 142432, Russia; roz_baz@mail.ru (R.K.B.); nd_golubeva@mail.ru (N.D.G.)
[3] Department of New Materials, National University of Science and Technology "MISIS", Leninskiy Prospect, 4, Moscow 119049, Russia; karpenkov.dy@misis.ru
[4] Department of Physics, Lomonosov Moscow State University, Leninskie Gory 1, Moscow 119991, Russia
[5] Joint Institute for High Temperature of the RAS, Moscow 125412, Russia; eagolovkova@gmail.com (E.A.G.); degtyarenkopn@gmail.com (P.N.D.)
[6] Depatment of Chemistry, Southern Federal University, B. Sadovaya Str., 105/42, Rostov-on-Don 344006, Russia; ieuflyand@sfedu.ru
* Correspondence: k_kamila@mail.ru (K.A.K.); dzhardim@icp.ac.ru (G.I.D.); Tel.: +7-(496)5227763 (G.I.D.)

**Abstract:** Bimetallic FeCo and FeNi nanoparticles attract much attention due to their promising magnetic properties and a wide range of practical applications as recording and storage media, catalytic systems in fuel cells, supercapacitors, lithium batteries, etc. In this paper, we propose an original approach to the preparation of FeCo- and FeNi/N-doped carbon nanocomposites by means of a coupled process of frontal polymerization and thermolysis of molecular co-crystallized acrylamide complexes. The phase composition, structure, and microstructure of the resulting nanocomposites are studied using XRD, IR spectroscopy, elemental and thermal analysis, and electron microscopy data. The main magnetic characteristics of the synthesized nanocomposites, including the field dependences and the ZFC-FC curves peculiarities, are studied. It is shown that the obtained FeCo/N-C nanocomposites exhibit exchange bias behavior at low temperatures. In turn, FeNi/N-C nanocomposites are ferromagnetically ordered.

**Keywords:** bimetallic nanoparticles; nanocomposites; thermolysis; magnetic properties

## 1. Introduction

Materials containing magnetic nanoparticles are important basic components for many applications such as high-density magnetic recording media [1–3], catalysts [4–7], contrast agents, hyperthermia therapy, and drug delivery systems in biomedicine [8–12]. Among these nanomaterials, FeCo and FeNi bimetallic nanoparticles are of particular interest because of their promising magnetic properties, which can be controlled depending on the synthesis method, composition, and ratio of system components [13–16]. It is known that Co atoms increase the coercive force and remanent magnetization of these nanomaterials with an increase in their concentration [17]. CoFe alloys have high saturation magnetization $M_s$ and Curie temperature ($T_c \approx 900$ °C), which makes it possible to consider them as high-temperature materials capable of operating in magnetic bearings of high-speed electric machines, flywheels of gas turbine engines, etc. [18]. FeNi alloys have similar characteristics and zero magnetostriction, as well as high anisotropic magnetic resistance [19]. To obtain the considered nanomaterials, various synthesis methods are used; for example, chemical coprecipitation and reduction thermal decomposition, polyol processes, sonochemical

decomposition, etc. [20–24]. In [25], ferromagnetic FeCo nanoparticles were obtained in a boiling ethylene glycol solution followed by annealing in a hydrogen flow at temperatures of 350, 400 and 600 °C. The modified polyol synthesis in the absence of surfactants was used to obtain FeCo nanoparticles with high magnetization (230–232 emu/g) with atomic composition $Fe_{60}Co_{40}$ [14,21]. To obtain FeCo and FeNi systems, methods of chemical reduction with sodium borohydride are widely used, which allow avoiding the formation of metal borides [26–28]. In most cases, synthesized FeCo and FeNi nanoparticles are characterized by a core–shell architecture. Such structure makes it possible to control the physical and chemical properties of the product based on the composition and structure of the core and shell [26]. The shell can be of oxide, carbide, carbon, or other nature and, on the one hand, protect metals from oxidation and, on the other hand, affect the maximum metal magnetization, reducing the latter [29]. Spherical air-resistant bimetallic magnetic FeNi and FeCo nanoparticles with an average diameter of 15 nm and a core–shell structure were synthesized by reduction using sodium borohydride [26]. The core of the nanoparticles consists of FeNi or FeCo alloys, and the ferrihydrite shell prevents oxidative degradation of nanoparticles. To prevent aggregation of the resulting nanoparticles, they are usually synthesized in the presence of stabilizers or by post modification of the surface of nanoparticles by adsorption of long-chain organic molecules [13,30–32]. For example, in [33], bimetallic FeNi nanoparticles stabilized by an amphiphilic block copolymer polystyrene-block-poly(acrylic acid) were obtained by the reduction in $FeSO_4 \cdot 7H_2O$ with potassium borohydride in the presence of a surfactant.

It should be noted that metal nanoparticles encapsulated in a carbon shell (metal–carbon nanocomposites) are of great interest in connection with the prospects for their application in the creation of new materials for technology and medicine [34,35]. The inert carbon coating of the particles makes it possible to use them in medicine as biocompatible and non-toxic delivery vehicles for the diagnosis and therapy [36–39]. In [40], a high catalytic activity of metal–carbon nanocomposites was found, probably due to the structural and electronic state of the carbon shell and metal core [41,42].

Element-doped nanocomposites, in particular N-doped metal–carbon nanocomposites, are of particular interest, including for catalytic purposes [30,43–46]. It has been shown that bimetallic FeNi nanoparticles of $Fe_{20}Ni_{80}$ composition exhibit high activity in oxygen evolution reactions (OER) in alkaline electrolytes [5]. However, in terms of stability, they are inferior to catalysts of the following composition: $Fe_{50}Ni_{50}$ and $Fe_{80}Ni_{20}$. Therefore, the stability of catalysts is a critical factor in their development. At the same time, $Fe_{50}Ni_{50}$ nanoparticles have a high catalytic activity in OER at higher current densities, which makes this compound suitable for electrolysis conditions. The development of bimetallic nanoparticles of the type under consideration is of general importance for the creation of efficient heterogeneous catalysts. In [47], Ni nanoparticles modified with nitrogen-doped graphene layers active in hydrogenation reactions were reported. FeNi alloy nanoparticles encapsulated in nitrogen-doped carbon nanotubes showed high electrocatalytic activity in the redox reaction as a counter-electrode in solar cells [46]. The authors suggested that this circumstance is associated with the modification of the electronic properties of the carbon surface by encapsulated metal alloy nanoparticles.

Nitrogen-doped carbon or metal–carbon nanocomposites are usually synthesized by co-pyrolysis of precursors containing carbon and nitrogen. The choice of suitable precursors and the optimal pyrolysis temperature is a challenging task to obtain nanocomposites with high nitrogen content, high surface area and carbon with high graphite content and improved performance.

We proposed an original procedure for the preparation of metal/N-doped carbon nanocomposites using the coupled process of frontal polymerization (FP) and controlled thermolysis of acrylamide complexes of metal nitrates [48–52]. It should be noted that the FP of acrylamide complexes of transition and noble metal nitrates is the first case of purely thermal initiation of the process in the absence of material initiators described in the literature; the latter are widely used in the case of many other systems in the production of

polymeric materials and composites by the FP method [53–56]. The subsequent controlled thermolysis of the resulting metallopolymers or the controlled transition of FP to the combustion mode leads to the formation of metal–polymer nanocomposites. Previously, we synthesized monometallic M@N-doped carbon nanocomposites with a core–shell structure based on Co(II), Ni(II), Fe(III) acrylamide complexes as precursors, in which the carbonized polymer shell protects the metal core from oxidation and aggregation [57,58].

The advantage of the proposed original approach for obtaining bimetallic nanoparticles in an N-doped carbon matrix in this work is that cocrystalizates of acrylamide complexes of metal nitrates are used as a single-source precursor that combines the necessary elements in one molecule and can form a metallopolymer and a nanocomposite based on it in the frontal mode. Moreover, the formation of a metal–polymer intermediate, as shown earlier [48,57,59], promotes the formation of monodisperse nanoparticles homogeneously distributed in a carbonized polymer matrix. This work is aimed at obtaining FeCo- and FeNi/N-doped carbon nanocomposites by a coupled process of frontal polymerization and thermolysis of molecular co-crystallized Fe(III)/Co(II) and Fe(III)/Ni(II) acrylamide complexes, as well as the study of their composition, structure, and magnetic properties.

## 2. Materials and Methods

### 2.1. Reagents

Compounds $Fe(NO_3)_3 \cdot 9H_2O$ (Sigma-Aldrich, $\geq 98\%$, Chemie GmbH, Steinheim, Germany), $Co(NO_3)_2 \cdot 6H_2O$ (Sigma-Aldrich, 98%, Chemie GmbH, Steinheim, Germany), and acrylamide (AAm) (Sigma-Aldrich, $\geq 98\%$, Chemie GmbH, Steinheim, Germany) were used as initial components for obtaining bimetallic acrylamide complexes and nanocomposites based on them without additional purification. Benzene $C_6H_6$ (chemically pure, REAKHIM, Moscow, Russia) and diethyl ether $(C_2H_5)_2O$ (pure, REAKHIM, Moscow, Russia) were purified and distilled according to the standard procedure.

### 2.2. Synthetic Procedures

The general scheme for obtaining FeCo- and FeNi/N-doped carbon nanocomposites (FeCo/C-N and FeNi/C-N) can be represented as follows:

$$FeCo(Ni)AAm \xrightarrow[T_{in} = 130°C]{FP} FeCo(Ni)PolyAAm \xrightarrow[\substack{T_{ex} = 400 - 600°C, \\ SGA,\ 2 - 4\ h}]{Thermolysis} FeCo(Ni)/C - N$$

where SGA is self-generated atmosphere, $T_{in}$ is the temperature of the initiation of FP, $T_{ex}$ is the temperature of the thermolysis.

*Preparation of Fe(III)/Co(II)- (FeCoAAm) and Fe(III)/Ni(II) (FeNiAAm) monomeric acrylamide co-crystallized complexes.* A mixture of $Fe(NO_3)_3 \cdot 9H_2O$ and $Co(NO_3)_2 \cdot 6H_2O$ salts was mixed in a mortar in a ratio of 2:1 wt. % and the resulting mass was grinded with acrylamide ($CH_2=CHCONH_2$) in a ratio of 1:5 (mol/mol) to obtain a pasty mass. The resulting product was left in a desiccator over $P_2O_5$ for 10–12 h, then washed with benzene and diethyl ether, and dried in a vacuum at room temperature. The product yield was 94–98%. The elemental analysis data of the obtained monomeric complexes are presented in Table 1. The calculation of the elemental chemical composition of the compounds was carried out according to their gross formulas for compositions $Fe_3Co_2C_{36}H_{75}N_{18}O_{41}$ and $Fe_3Ni_2C_{36}H_{75}N_{18}O_{41}$; the experimental elemental composition of the compounds for C, H, and N was determined by organic microanalysis on a Vario EL cube elemental analyzer by burning the analyzed sample in the presence of an oxidizer in an inert gas flow. The content of Fe, Co, and Ni was determined by atomic absorption spectrometry on an AAS-3 instrument. During sample preparation, a weighed portion of the exact mass was dissolved using a mixture of three concentrated acids. IR spectra (tablet with KBr), $\nu/cm^{-1}$: 3190, 3290 (NH); 1665 (CO); 1385 ($NO_3$); 354 (M-O) for FeNiAAm; 3190, 3290 (NH); 1665 (CO); 1385 ($NO_3$); 354 (M-O) for FeCoAAm.

**Table 1.** Elemental composition of Fe(III)/Co(II) and Fe(III)/Ni(II) bimetallic acrylamide complexes.

| Composition | Found/Calculated, % | | | | |
|---|---|---|---|---|---|
| | C | H | N | Fe | Co (Ni) |
| $[Fe_3Co_2(CH_2CHCONH_2)_{12}(NO_3)_6(OH)_7(H_2O)_4]$ (FeCoAAm) | 26.13/25.41 | 5.36/4.44 | 15.09/14.82 | 10.51/9.84 | 6.78/6.93 |
| $[Fe_3Ni_2(CH_2CHCONH_2)_{12}(NO_3)_6(OH)_7(H_2O)_4]$ (FeNiAAm) | 25.48/25.42 | 5.26/4.44 | 15.47/14.82 | 10.23/9.84 | 7.22/6.92 |

*Preparation of FeCoPolyAAm and FeNiPolyAAm polymeric acrylamide co-crystallized complexes and their nanocomposites (FeCo/C-N and FeNi/C-N).* FeCoAAm and FeNiAAm powders were pressed into a cylindrical shape at a constant pressure of 50 bar, at room temperature of 20 °C for 5 min. As a result, cylindrical blanks with dimensions of 5 × 12 mm and a density of 1.45 g/cm³ were obtained. For FP, the resulting sample blanks were placed in a glass ampoule. The reaction wave was triggered by introducing a thermal perturbation into the lower part of the ampoule when it was immersed by 0.2 cm into the bath with a heat carrier (up to 15 s), after which the reaction proceeded in a self-sustaining mode.

The thermolysis of polymerized samples was carried out under static isothermal conditions at temperatures of 400 and 600 °C (in the case of FeNiPolyAAm) in a self-generated atmosphere for 2 and 4 h (preliminary evacuation of the studied samples was carried out at room temperature for 30 min).

### 2.3. Characterization

Elemental analysis of cocrystallized complexes and nanocomposite products for the content of C, H, N was carried out using a Vario EL cube elemental analyzer (Elementar GmbH, Hanau, Germany), and for Co, Ni, Fe, by atomic absorption spectroscopy (AAS-3, Zeiss, Jena, Germany). Monomeric and polymeric co-crystallizates and thermal transformation products were studied by IR spectroscopy in the range of 400–4000 cm⁻¹ using a Specord 75 IR spectrophotometer (Zeiss, Jena, Germany). Differential scanning calorimetry (DSC) and thermogravimetric analysis (TGA) were performed on Mettler Toledo DSC822e (Mettler Toledo, Greisensee, Switzerland) and TGA/SDTA851e (Mettler Toledo, Greisensee, Switzerland) instruments, respectively. X-ray diffraction of powders was studied on a DRON UM-2 diffractometer using CuKα radiation (λ = 1.54178 Å) (Moscow, Russia), decoding and calculation of the lattice parameters by the Rietveld method was carried out using a software package developed at the Department of Physical Materials Science, NRTU MISiS [60]. Spectra for samples with Fe-Ni and Fe-Co nanoparticles were recorded at a constant rate of 1 deg/min; the step was 0.005 degrees. For electron microscopic studies, a JEM 2100 transmission electron microscope (JEOL Ltd., Tokyo, Japan) with an accelerating voltage of 200 kV was used. For the study, the sample was dispersed in isopropyl alcohol using an ultrasonic bath for 5 min. The resulting solution was dropped onto a copper grid coated with an amorphous carbon film. For X-ray microanalysis of elements (EDX), an attachment to a TEM (an INCA ENERGY analytical system, OXFORD INSTRUMENTS, Oxford Instruments, Concord, UK) was used. The magnetic properties of the metal–polymer nanocomposites were measured using a Quantum Design Physical Properties Measurement System (PPMS-9) vibrating magnetometer (PPSM-9, Quantum Design, Zürich, Germany). Hysteresis loops were obtained in magnetic fields from 0 to 20,000 Oe in the temperature range from 10 to 300 K. The temperature dependences of the ZFC-FC (zero-field cooling–field–cooling) magnetization were measured in the same temperature range at an external magnetic field H = 500 Oe.

### 3. Results and Discussion

### 3.1. Composition and Structure of Bimetallic Nanocomposites

Synthetic methods that use two or more precursors to obtain nanostructured materials have certain limitations, since the formation of a nanocrystal often depends on the mutual

reactivity of these constituent components, their stability, etc. Many of these problems, including the use of toxic and volatile compounds at high temperatures, can be avoided if the precursors are molecular complexes, the so-called single source precursors that combine in one molecule both the corresponding metal ions and constituent elements, for example, N, P, S, Se, etc. [61]. To obtain FeCo/C-N and FeNi/C-N nanocomposites, co-crystallized acrylamide complexes of Fe(III)/Co(II) and Fe(III)/Ni(II) metal nitrates were used as molecular precursors, which were subjected to FP followed by thermolysis according to the scheme depicted in Figure 1 [62].

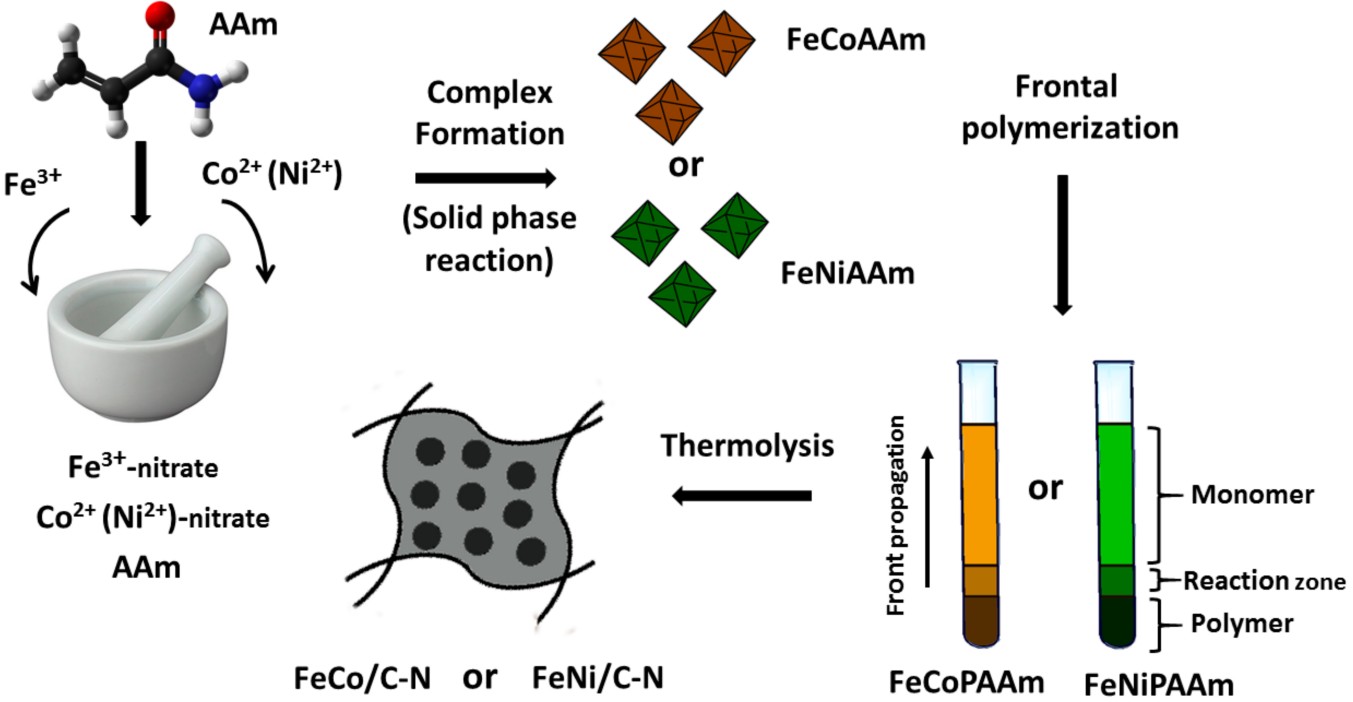

**Figure 1.** A schematic route for the preparation of FeCo/C-N and FeNi/C-N nanocomposites.

This approach, as we previously showed [48–52], makes it possible to obtain matrix-stabilized and highly dispersed metal nanoparticles with controlled size and homogeneous distribution in the matrix space, which is of considerable interest for various practical applications. The presence of a carbonized polymer shell ensures the chemical stability of nanoparticles and prevents them from aggregation. Both the temperature and the reaction time proved to be optimal for the decomposition of the intermediate metallopolymer.

### 3.1.1. IR Spectroscopy

As a result of dehydration during thermal transformations of cocrystallizates, the absorption bands associated with the modes of crystallization water $\nu$(O–H) 3000–3600 cm$^{-1}$, $\rho_\omega$(O-H) + $\nu$(M-OH$_2$) 880 cm$^{-1}$ [63] disappear. At the same time, the intensity of the modes caused by $\delta$(O-H) + $\nu$(C=C) 1616 cm$^{-1}$, $\delta$(CH$_2$) + $\delta$(M-OH$_2$) 582 cm$^{-1}$ decreases. During polymerization, the IR absorption spectra of the dehydrated monomer undergo changes associated with a decrease in the intensity of the absorption band of the valence modes of the C=C bond and convergence of the absorption frequencies of the valence modes of the C=O bond, which leads to the appearance of one broadened absorption band in the region of 1390–1563 cm$^{-1}$. The spectrum of polymer complexes retains absorption bands in the range of OH and NH stretching vibrations (3415–3419 cm$^{-1}$), $\nu$NO$_3$ (1380 cm$^{-1}$). The absence of absorption bands $\nu$(C=C) and $\delta$(=CH) at 1580 and 980 cm$^{-1}$ confirms the consumption of C=C bonds during polymerization. In the low-frequency region, oscillations of large-amplitude protons in the region of 720–520 cm$^{-1}$ are also preserved (Figure 2).

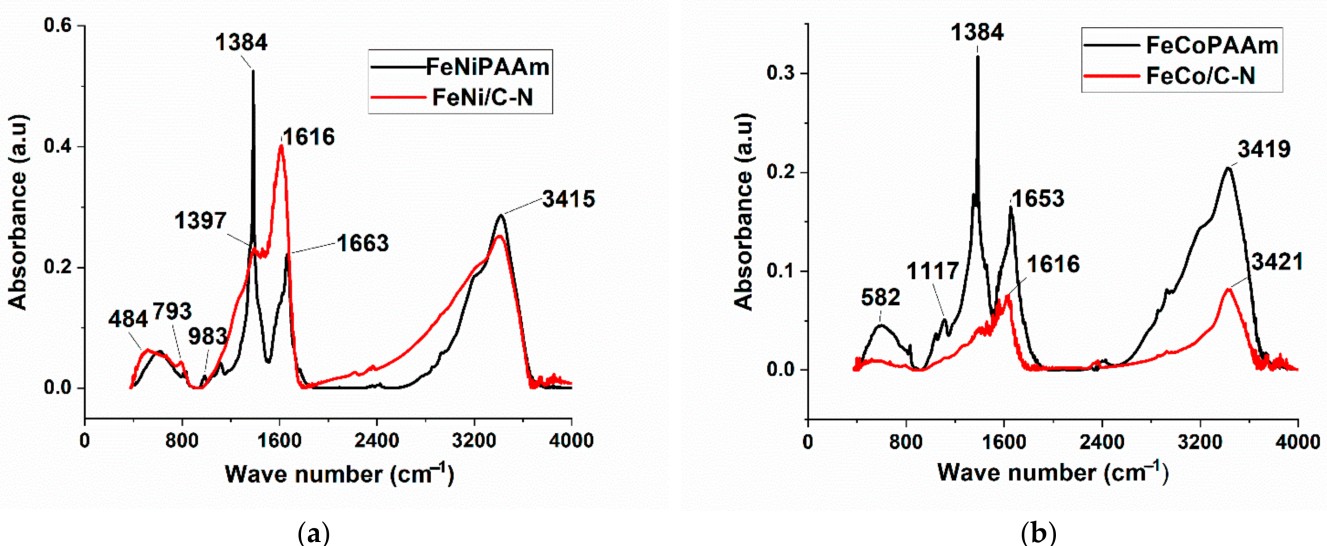

**Figure 2.** IR spectra of FeNiPAAm (**a**), FeCoPAAm (**b**) polymer complexes and their nanocomposites FeNi/C-N (**a**) and FeCo/C-N) (**b**) (1% KBr tablets).

It can be seen from Figure 2 that during thermolysis, the initial structure of the three-dimensional polymer complex is destroyed with the preservation of individual fragments, as evidenced by the bands of OH and NH stretching vibrations (3415–3420 cm$^{-1}$), the out-of-plane bending vibrations of NH$_2$ (900 cm$^{-1}$) and low-frequency vibrations (450–900 cm$^{-1}$). In the region of 1600 cm$^{-1}$, broad absorption bands appear, indicating the presence of C=C stretching vibrations characteristic of conjugated systems, such as triene and diene fragments. All absorption bands have a Gaussian shape, which indicates that interatomic and deformation vibrations occur in a limited space, i.e., the system is characterized by a high degree of crosslinking.

### 3.1.2. Thermal Analysis

To reveal the temperature regions of the main processes during thermal transformations, the initial acrylamide co-crystallized complexes were studied by thermal analysis methods. The nature of the DSC (Figure 3a) and TGA (Figure 3b,c) curves recorded for the original co-crystallized complexes showed that the thermal effects of polymerization and pyrolysis (thermal decomposition of the polymer structure) overlap: the beginning and end are between 130 and 300 °C and, in general, indicate the complex nature of the ongoing conjugated processes (Figure 3). Endothermic peaks in the range of 50–100 °C are associated with dehydration reactions, which is consistent with the weight loss data on the TGA curves (Figure 3b,c), corresponding to ~6 mol H$_2$O in the case of FeNiAAm and up to 10 mol H$_2$O for FeCoAAm.

Intense weight loss of 30–35 wt.% in the region of 150–350 °C is associated with the processes of decarboxylation and breaking of bonds in the main carbon chain of the polymer matrix, accompanied by cyclization processes, etc.

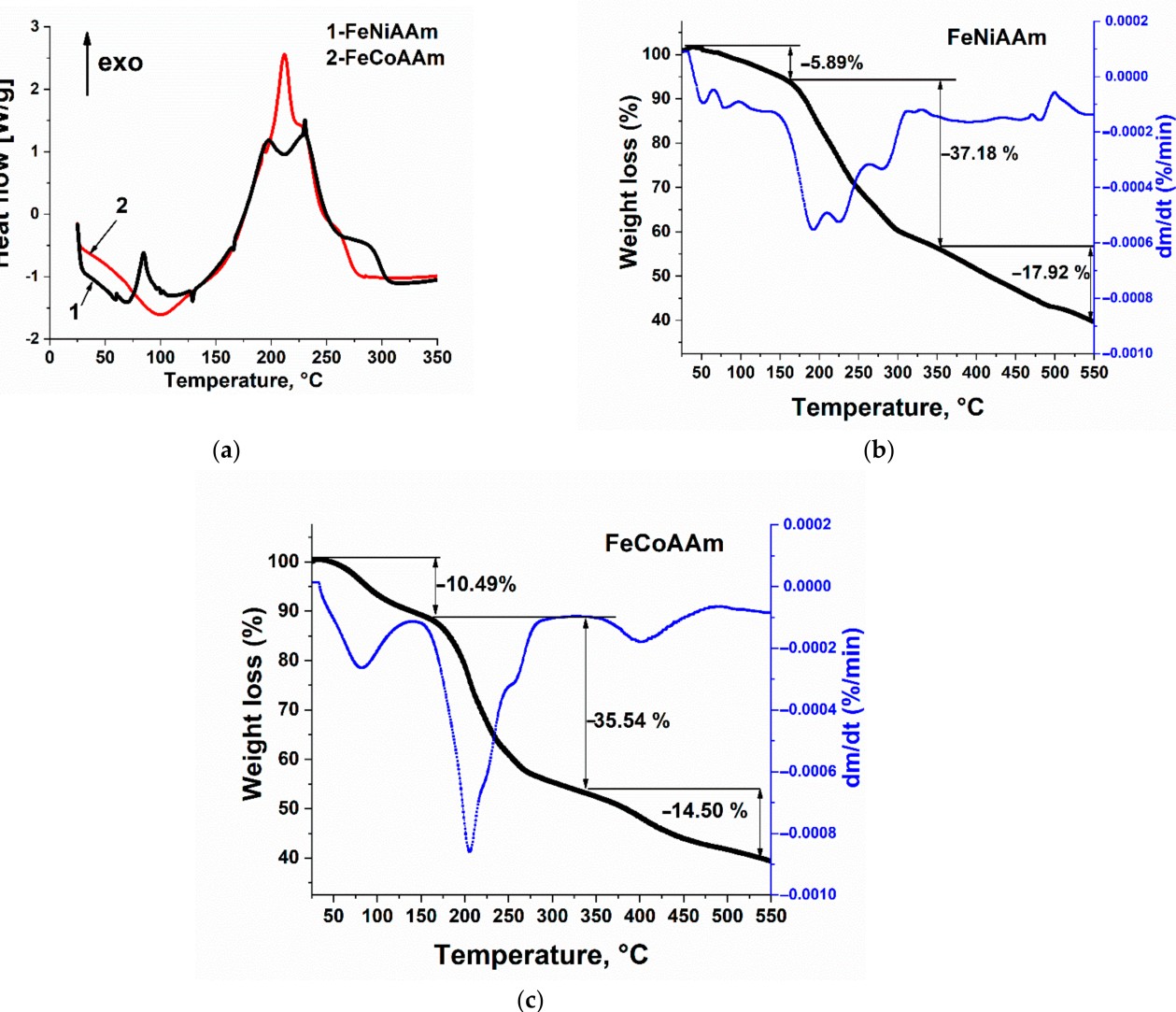

**Figure 3.** DSC (**a**) and TGA curves (**b**,**c**) of cocrystallized acrylamide complexes FeNiAAm (**b**) and FeCoAAm (**c**): 1—FeNiAAm; 2—FeCoAAm.

### 3.2. Phase Composition and Microstructure of FeCo/C-N and FeNi/C-N Nanocomposites

### 3.2.1. X-ray Diffraction

According to the equilibrium diagram of the Fe-Co (2:1) system, in the region with a given ratio of iron and cobalt, an iron-based substitutional solid solution with the Pm-3m structure (CsCl structural type) is formed.

Figure 4 shows the X-ray diffraction pattern of the FeCo/C-N nanocomposite obtained at 400 °C. All peaks in the diffraction pattern are identified by this structure, the crystal lattice parameter a = 2.8569 ± 0.0008 Å, the broad diffraction peak (halo) at 2θ = 23° can be attributed to the carbonized polymer shell. According to the Scherer equation D = Kλ/β cosθ [64], the size of the coherent scattering block is D = 105 ± 2 Å.

When the ratio of iron and nickel is 2:1, two phases are formed according to the equilibrium diagram; an iron-based solid solution with the Im3m structure and a $Fe_3Ni$ phase with the Pm-3m structure (structural type L1$_2$). However, due to the very close lattice parameters of the $Fe_3Ni$, FeNi, and $Ni_3Fe$ phases and the Fe-based high-temperature solid solution with the Fm3m structure (according to the literature, Δa = 0.0003 nm), it is impossible to distinguish between these phases using X-ray diffraction analysis.

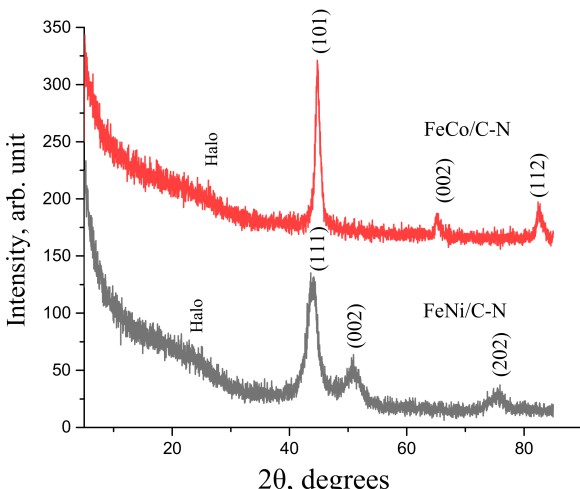

**Figure 4.** X-ray diffraction patterns of FeCo/C-N and FeNi/C-N nanocomposites obtained at 400 and 600 °C, respectively.

Figure 4 shows the X-ray diffraction pattern of the FeNi/C-N nanocomposite obtained at 600 °C; all peaks in the diffraction pattern are identified with the Fm-3m phase with a lattice parameter a = 3.588 ± 0.004 Å; the size of the coherent scattering block is D = 50 ± 1 Å. The broad diffraction peak (halo) at 2θ = 23° can be attributed to the carbonized polymer shell.

### 3.2.2. Electron Microscopy Studies

The EDX analysis of the samples confirms the presence of FeCo and FeNi bimetallic nanoparticles in the matrix (Figure 5), which also agrees with the data of elemental microanalysis (Table 2).

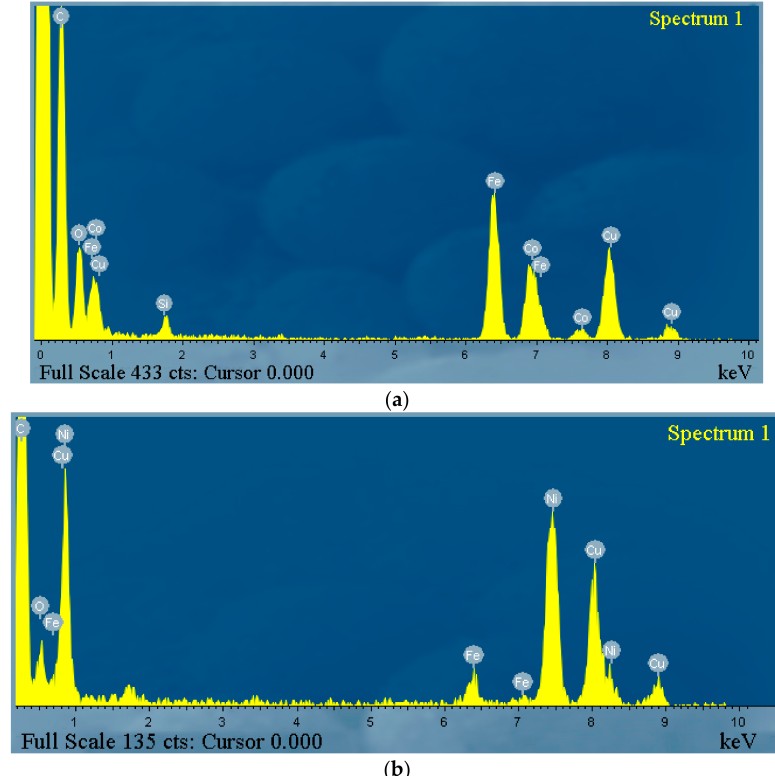

**Figure 5.** EDX spectra of FeCo/C-N (**a**) and FeNi/C-N (**b**) nanocomposites obtained at 400 (**a**) and 600 °C (**b**).

**Table 2.** Elemental analysis data for FeCo/C-N and FeNi/C-N nanocomposites.

| Samples | Elemental Composition, wt. % | | | | |
|---|---|---|---|---|---|
| | **C** | **H** | **N** | **Fe** | **Co (Ni)** |
| FeCo/C-N-400*-2 | 35.53 ± 0.23 | 3.24 ± 0.02 | 8.97 ± 0.14 | 13.2 ± 0.66 | 6.24 ± 0.13 |
| FeNi/C-N-400-4 | 37.37 ± 0.27 | 2.51 ± 0.01 | 10.14 ± 0.04 | 14.50 ± 0.72 | 7.53 ± 0.37 |
| FeNi/C-N-600-4 | 46.54 ± 0.17 | 1.52 ± 0.03 | 7.53 ± 0.08 | 18.4 ± 0.92 | 9.6 ± 0.48 |

\* The numerals in the designation of the samples indicate the thermolysis temperatures of 400 and 600 °C, respectively, the last digits indicate the thermolysis time of 2 and 4 h, respectively.

The ratio of the Fe and Co(Ni) peaks in the nanocomposites according to the EDX data corresponds to the molecular ratio of the precursor used in the synthesis procedure. In addition, the amount of oxygen due to the formation of oxides is rather low, which is consistent with the data of X-ray diffraction patterns, according to which no oxide phases were found in FeCo/C-N and FeNi/C-N nanocomposites.

Figures 6 and 7 show TEM images of the resulting bimetallic nanocomposites. Histograms of particle size distribution are calculated based on the results of TEM image analysis using Image J software, version 1.53t.

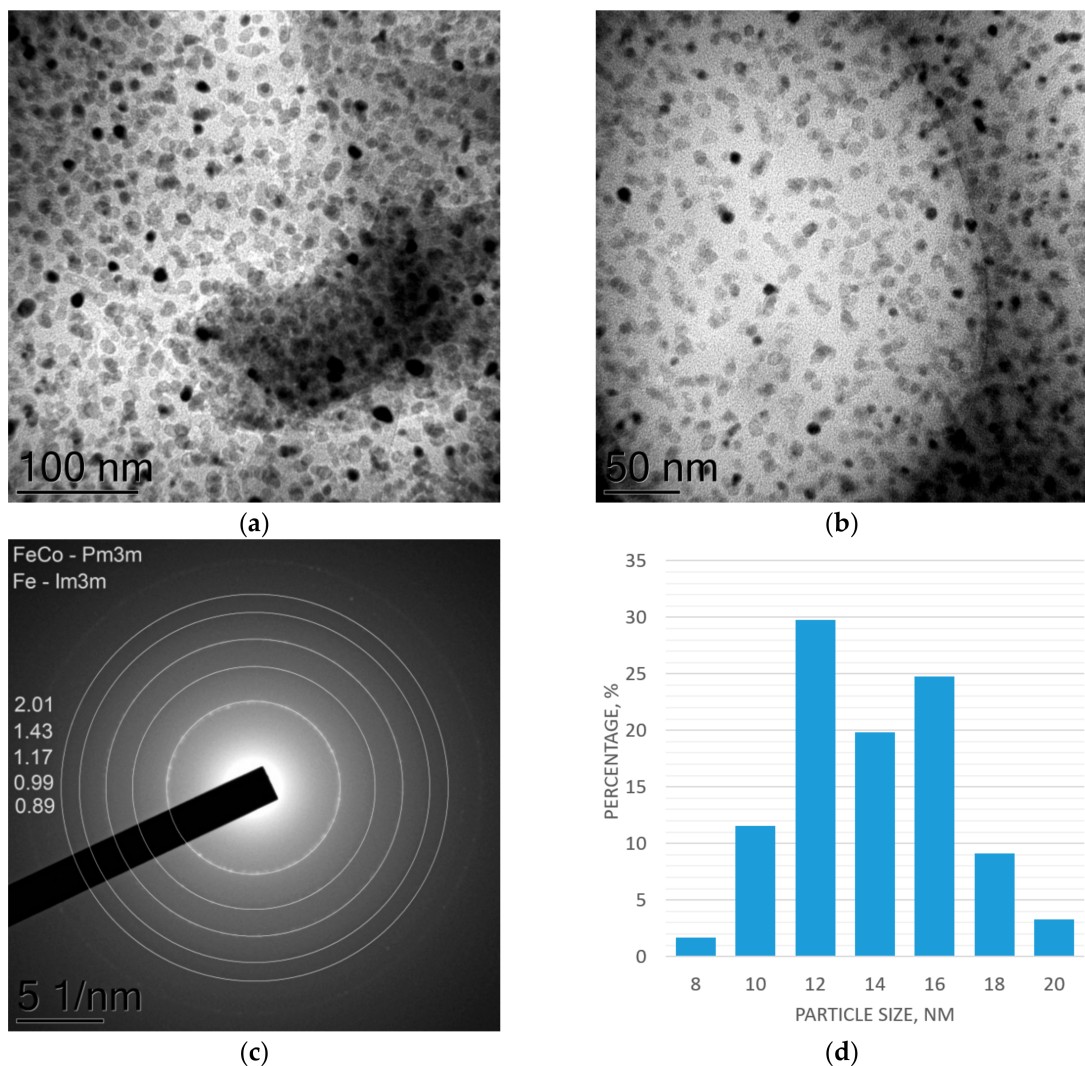

**Figure 6.** TEM results of the FeCo/C-N sample. (**a**,**b**) TEM image; (**c**) the SAED diffraction pattern; (**d**) size distribution histogram.

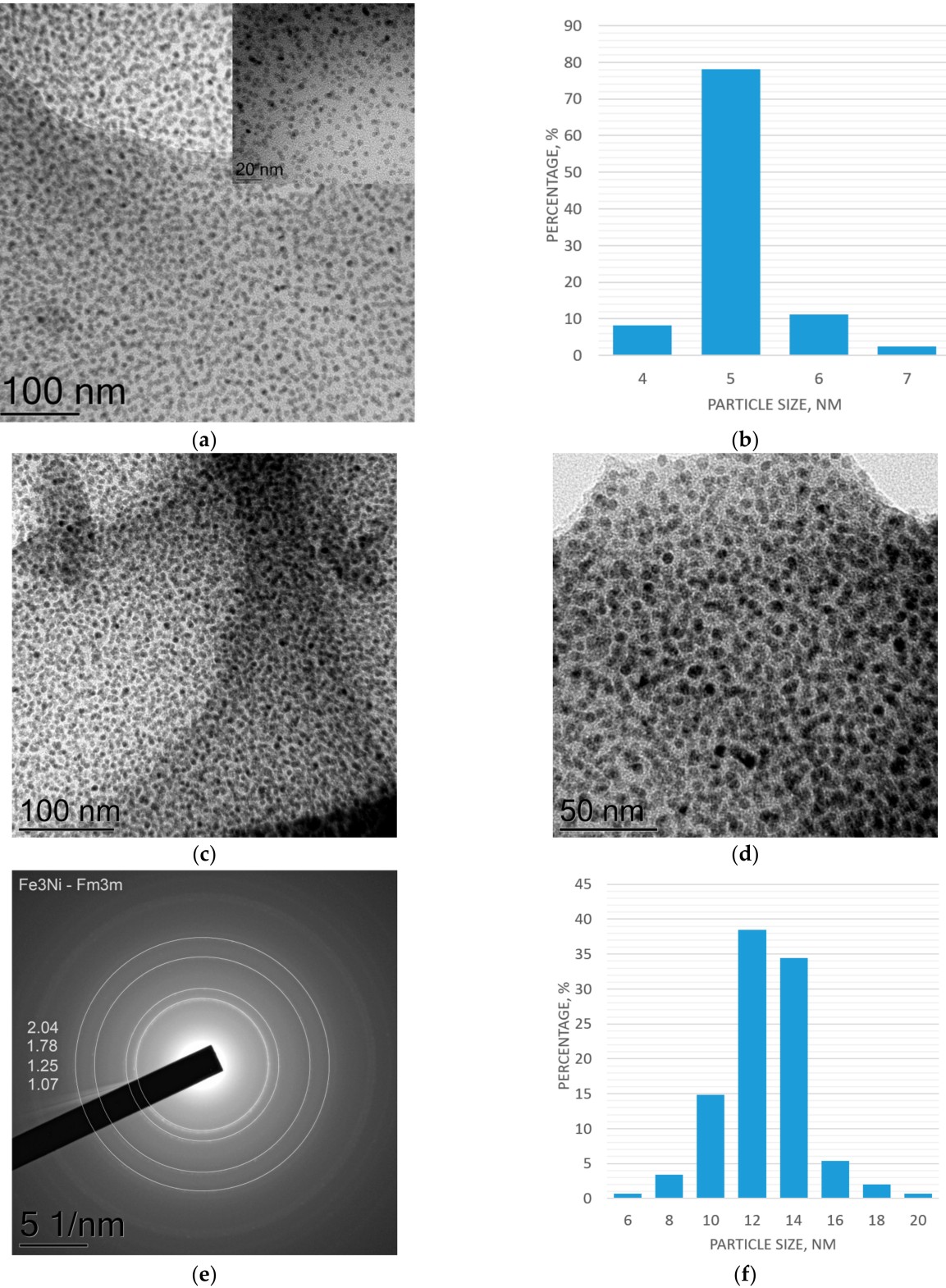

**Figure 7.** TEM results of the FeNi/C-N-400-4 sample: (**a**) TEM image; (**b**) size distribution histogram. TEM results of the FeNi/C-N-600-4 sample: (**c**,**d**) TEM image; (**e**) the SAED diffraction pattern; (**f**) size distribution histogram.

According to TEM data, the samples have a similar microstructure morphology, consisting of predominantly spherical bimetallic nanoparticles uniformly distributed in an N-doped carbon matrix. It can be seen from the histograms that FeNi nanoparticles have

a higher degree of monodispersity compared to FeCo particles. There is also a tendency to form chain-like structures. Considering the latter, as well as the denser nature of the distribution of FeNi nanoparticles in the matrix in comparison with FeCo nanoparticles, it is of interest to further evaluate their electrically conductive properties [65]. As expected, an increase in the synthesis temperature from 400 to 600 °C, as shown in the case of FeNi/C-N (Figure 7), leads to an increase in the size of nanoparticles and a broadening of their size distribution.

It is interesting to note that in the case of a monometallic system, for example, an acrylamide complex of Fe(III) nitrate, under similar conditions, the formed Fe/C-N composite consists of irregularly shaped porous particles ranging in size from 50 to 500 nm, and the powder particles contain $Fe_3C$ nanocrystals, the average size of which is in the range from 10 to 80 nm [52].

The SAED analysis in the case of FeCo/C-N confirms the formation of a FeCo bimetallic alloy; in addition, a metallic iron phase $\alpha$-Fe (sp. gr. Im3m) is detected, while for FeNi/C-N nanocomposites, SAED indicates the presence of a $Fe_3Ni$ phase with the Fm-3m structure. Due to the insufficient number of observed reflections and their low intensity, it is difficult to accurately identify the specific intermetallic phase (or phases) of the FeNi/C-N nanocomposite at this stage.

### 3.3. Magnetic Properties of FeCo/C-N and FeNi/C-N Nanocomposites

The field dependences of the magnetization of the obtained nanocomposites at 10 and 300 K are shown in Figure 8. The main parameters of the hysteresis loop are presented in Table 3. It is worth noting that the magnetic moment of the samples is not normalized on the mass of metal nanoparticles. The nature of the hysteresis loop at 300 K has the typical behavior of a material with a ferromagnetic order. At low temperatures, a significant change in the shape of the loop is observed for FeCo/C-N; in particular, a continuous increase in the magnetization is observed in high fields. Along with the lack of saturation observed in low-temperature loops, the hysteresis loops for FeCo/C-N nanocomposites are shifted towards a negative magnetic field. This is due to the presence of an exchange bias field of 1061 Oe, which usually arises at the interface between the ferro- and antiferromagnetic phases [66]. Thus, the formation of a mixed oxide on the surface of metal particles with antiferromagnetic ordering is assumed. In this case, the absence of an exchange bias field at room temperature is due to the low values of the Curie temperature for these oxides.

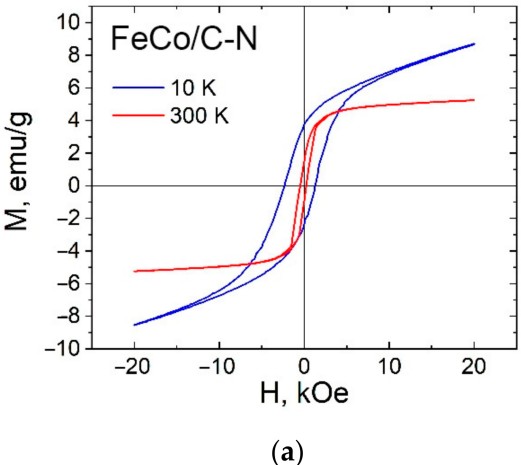

(**a**)

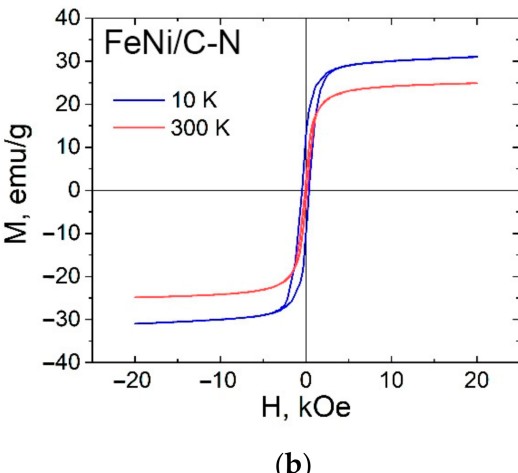

(**b**)

**Figure 8.** Hysteresis loops for FeCo/C-N (**a**) and FeNi/C-N (**b**) nanocomposite samples at T = 10 and 300 K.

The field dependence of FeCo/C-N is explained by the presence of uncompensated interfacial spins, which experience competition between the exchange and Zeeman energies, along with the well-known surface anisotropy from the magnetic nanoparticles responsible

for the slow approach to saturation. This magnetic behavior, along with the AFM behavior of the mixed oxide, is also responsible for the lack of saturation observed in all low-temperature loops [67,68].

**Table 3.** Magnetic characteristics of FeCo/C-N and FeNi/C-N nanocomposites.

| Sample | $M_s$, emu/g (A m$^2$ kg$^{-1}$) | | $M_r$, emu/g (A m$^2$ kg$^{-1}$) | | $M_r/M_s$ | | $H_c$, Oe (kA m$^{-1}$) | |
|---|---|---|---|---|---|---|---|---|
| | 10 K | 300 K | 10 K | 300 K | 10 K | 300K | 10 K | 300 K |
| FeCo/C-N-400 | 8.70 | 5.01 | 3.84 | 1.74 | 0.44 | 0.34 | 1900 (152) 839 (67.1) | 364 (29.1) |
| FeNi/C-N-600 | 30.08 | 24.01 | 13.02 | 1.93 | 0.43 | 0.15 | 489 (39.12) | 100 (8) |

The zero-field and field-cooled (ZFC-FC) magnetization measurements under an applied field of 500 Oe for FeCo/C-N and FeNi/C-N nanocomposites (Figure 9) exhibit a blocking process typical of the assembly of very weakly interacting single-domain magnetic particles with a distribution of blocking temperatures [69–71].

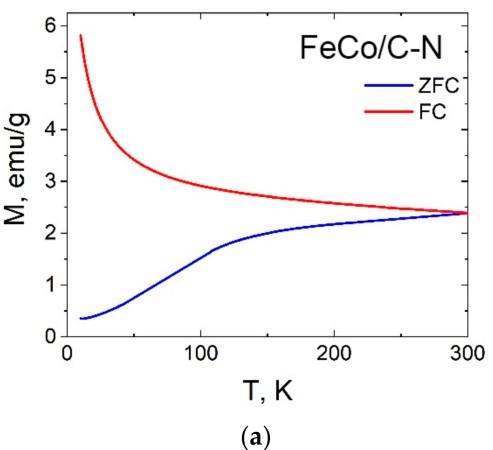

(**a**)

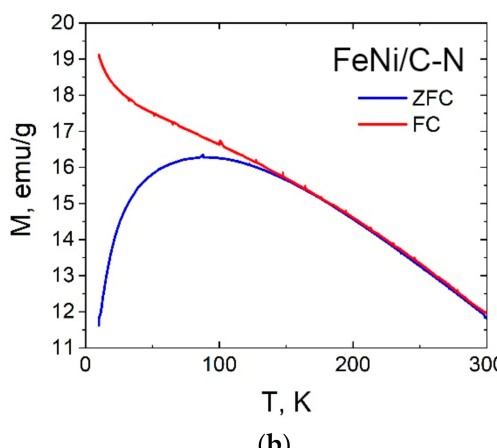

(**b**)

**Figure 9.** Dependence of magnetization on temperature in the ZFC-FC modes for FeCo/C-N (**a**) and FeNi/C-N (**b**) nanocomposite samples.

The ZFC magnetization curves have a maximum at temperature ($T_{max}$), which is related to the average blocking temperature ($T_B \propto bT_{max}$), where b is a proportionality constant depending on the type of size distribution [72]. To determine blocking $T_B$, the first derivative of difference $M_{ZFC}(T) - M_{FC}(T)$ versus the temperature is calculated. The mean $T_B$ is defined as the temperature at which the integral value (area under the d($M_{ZFC}(T) - M_{FC}(T))$/dT)) after appropriate normalization is 50%. Although d($M_{ZFC} - M_{FC}$)/dT does not reach zero at low temperatures, the TB trend is consistent with the trend of $T_{max}$. The values of $T_{max}$ and $T_B$ decrease with decreasing particle size. The shift of $T_{max}$ from $T_B$ and, as a result, the value of the b coefficient is determined by the particle size distribution. In our case, the value of b remains constant and amounts to 4.3 for the FeCo/C-N nanocomposites and to 4.1 for the FeNi/C-N nanocomposites (Table 4).

**Table 4.** Magnetic characteristics of FeCo/C-N and FeNi/C-N nanocomposites.

| Sample | $T_{irr}$, K | $T_{max}$, K | $T_B$, K | b |
|---|---|---|---|---|
| FeCo/C-N-400-2 | 300 | 300 | 73 | 4.1 |
| FeNi/C-N-600-4 | 124 | 86 | 20 | 4.3 |

The temperature below which the ZFC and FC curves exhibit an irreversible character ($T_{irr}$) is associated with the blocking of the largest particles, assuming that the anisotropy energy barrier is determined by the magnetocrystalline anisotropy [73].

It should be noted that FeNi/C-N nanocomposites, as noted above, are characterized by a higher dispersion of the metal phase and a uniform distribution in the matrix space than FeCo/C-N nanocomposites. This affects the nature of the ZFC/FC curves. In particular, the characteristics of FeCo/C-N nanocomposites, such as high $T_{irr}$ values, arw compatible with a large average size and a broad size distribution of these magnetic nanoparticles, which is consistent with the TEM images.

In the nanocomposites, the confluence of the FC and ZFC curves immediately follows the maximum of the ZFC curve, which is associated with the degree of aggregation of bare metal nanoparticles [74], which, in turn, decreases significantly when polymers are mixed. This is confirmed by the TEM images showing a uniform distribution of well-separated particles.

The continuous increase in MFC with decreasing temperature indicates that interparticle interactions, if present, are weak [73].

## 4. Conclusions

We successfully confirmed the possibility of obtaining magnetically active nanoparticles of a given composition and stabilized by an N-doped carbonized polymer matrix, which are formed simultaneously during the frontal polymerization of molecular precursors and their subsequent controlled thermolysis. Although this original method for the preparation of metal–polymer nanocomposites was proposed by us earlier, as discussed above, using the example of mono- and bimetallic molecular precursors, in this work, this approach was developed considering the variation of both the time and temperature of synthesis. Detailed structural studies involving XRD, TEM, SAED and EDX data made it possible to establish the phase composition of nanocomposites: in the case of the FeCo/C-N nanocomposite, phases of the substitutional solid solution of the FeCo bimetallic alloy (Pm3m structure) were identified; for FeNi/C-N, it was the $Fe_3Ni$ phase with the Fm-3m structure. Such structural analysis for these systems was carried out for the first time. The advantage of the proposed approach for obtaining nanocomposites of the type under consideration also presents the possibility of homogeneous dispersion of the formed nanoparticles in an N-doped carbonized polymer matrix, which is confirmed by TEM and EDX data. An increase in the synthesis temperature leads to an increase in the size of nanoparticles and a broadening of their size distribution. For the first time, the magnetic properties of the systems under consideration, including the ZFC and FC modes, were studied. It was shown that the magnetic behavior of the FeCo/C-N nanocomposites is not trivial and manifests the presence of a thin layer of antiferromagnetic oxides on the surface of each particle.

Thus, the advantage of the proposed synthesis of bimetallic nanoparticles based on the frontal polymerization of molecular precursors followed by controlled thermolysis is not only that this method can be used to obtain stabilized magnetic nanoparticles resistant to oxidation and aggregation; the considered approach allows effective control of the size of nanoparticles and, ultimately, their magnetic properties. It should also be noted that in order to obtain nanocomposites of the type under consideration with desired magnetic characteristics, it is also important to optimize the ratio between the components of the bimetallic alloy, which we plan to present in subsequent publications.

**Author Contributions:** Conceptualization, K.A.K. and G.I.D.; methodology, I.E.U.; software, L.S.B.; investigation, G.D.K., R.K.B., N.D.G., D.Y.K., E.A.G. and P.N.D.; resources, D.Y.K., P.N.D. and E.A.G.; writing—original draft preparation, K.A.K. and G.I.D.; writing—review and editing, G.I.D., I.E.U., D.Y.K. and E.A.G.; visualization, R.K.B., D.Y.K. and E.A.G. All authors have read and agreed to the published version of the manuscript.

**Funding:** This research received no external funding.

**Institutional Review Board Statement:** Not applicable.

**Informed Consent Statement:** Not applicable.

**Data Availability Statement:** Not applicable.

**Acknowledgments:** This work has been carried out in accordance with the state tasks, state registration No. AAAA-A19-119041090087-4 and AAAAA19119032690060-9 using the equipment of the Federal Research Center of Problems of Chemical Physics and Medicinal Chemistry RAS as well L.S.B. performed analysis of XRD data with the financial support of the state project of the Ministry of Education and Science of the Russian Federation "Project code FSFF-2023-0007". Magnetic experiments at helium temperature were obtained using equipment of the Lebedev Physical Institute's Shared Facility Center.

**Conflicts of Interest:** The authors declare no conflict of interest.

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
