# Peer review of "Polymer-Assisted Synthesis, Structure and Magnetic Properties of Bimetallic FeCo- and FeNi/N-Doped Carbon Nanocomposites"

_magnetochemistry, doi:10.3390/magnetochemistry9100213_

Round 1

Reviewer 1 Report

I lack a clear statement of the novelty: what is new according to your cited previous paper? What is application potential for these NPs? The Introduction should not be so long and detailed, but the novelty statement should be presented clearly.

Besides the structural data, You presented the results of the magnetic measurements I have an expertise. Again I lack a clear statement what new information the magnetic measurements gave. For instance, below 20 K the paramagnetic "tail" always appear. It is evidently more visible on a weak main background; moreover, it is higher for the defected structures having uncompensated charges/spins. You mentioned an exchange bias field together with a commonly accepted physical reason. But you did not evaluate it, did not explain a reason for your NPs and did not cite any paper in this field. What mass was taken for the normalization of the magnetic moment in Fig. 7? Is any correlation of the magnetic parameters, e.g., remanence and coercivity, with the similar bulk alloys? The coercive field is relatively high. The results of the ZFC-FC measurements should be also explain better with the corresponding citations. I generally suspect the theory taking an integral from the derivative of the difference. You mentioned a usual reason of magnetic anisotropy, but did not explain well a relation to the studied NPs. Why did you not perform the measurements above room temperature for the Fe-Co NPs?  

You have to reread the manuscript before the submission. There is a Russian text in lines 87 and  140-1 (what is the subject in the latter sentence?).  English should be checked. There are some parts with strange formulations and gramma mistakes: e.g., line 41 "ratio o " should be "of"; line 45 "able capable". Think also about the readability of your text, e.g., there should be some introductory sentence as "Figure 1 presents" in line 231.

The conclusion is out of my understanding, it should be fully rewritten. However, it is understandable in case when the novelty and motivations are unclear. 

The figures should be formatted the SAME way. You used different colors in similar Figs. 1(a,b) and 2(b,c). The latter is important because you used the second y-axis of the red color. The figure captions are not informatory enough, e.g., you forget to put the annealing temperature to Fig. 3(a). I recommend to add labels of the NPs for each figure as in Figs. 7-8.

I also recommend to reorganize the text structure slightly. The Introduction is full of technical details you could move to the second material section. For the Material section, two subsections (synthesis and characterization) would be quite enough. Think to move subsections 3.1-3.3 also to the Material-synthesis or present separately. I lack this separation between the synthesis, experiments performed to check the synthesis and the results obtained for the synthesized NPs.  

Author Response

Dear Reviewer,

We greatly appreciate your thoughtful comments that helped improve the manuscript.

Thank you very much for your effort.

In the following, we give a point-by-point reply to your comments:

Comments and Suggestions for Authors

I lack a clear statement of the novelty: what is new according to your cited previous paper? What is application potential for these NPs? The Introduction should not be so long and detailed, but the novelty statement should be presented clearly.

Besides the structural data, You presented the results of the magnetic measurements I have an expertise. Again I lack a clear statement what new information the magnetic measurements gave. For instance, below 20 K the paramagnetic "tail" always appear. It is evidently more visible on a weak main background; moreover, it is higher for the defected structures having uncompensated charges/spins. You mentioned an exchange bias field together with a commonly accepted physical reason. But you did not evaluate it, did not explain a reason for your NPs and did not cite any paper in this field. What mass was taken for the normalization of the magnetic moment in Fig. 7? Is any correlation of the magnetic parameters, e.g., remanence and coercivity, with the similar bulk alloys? The coercive field is relatively high. The results of the ZFC-FC measurements should be also explain better with the corresponding citations. I generally suspect the theory taking an integral from the derivative of the difference. You mentioned a usual reason of magnetic anisotropy, but did not explain well a relation to the studied NPs. Why did you not perform the measurements above room temperature for the Fe-Co NPs?  

Response: The main goal of the article is to explore a new approach for the synthesis 3d metal-based N-doped nanocomposites. An investigation of their magnetic properties allows us to evaluate their possible practical application. In addition, magnetic characteristics serve as a marker of particle size and structure peculiarities. As shown, the magnetic behavior of the FeCo/C-N nanocomposites is not trivial and manifests the presence of a thin layer of antiferromagnetic oxides on the surface of each particle. The latter cannot be detected by means of XRD.

As mentioned in the text of the article, we have developed an original method for obtaining mono- and bimetallic nanoparticles by combining the polymerization process and controlled thermolysis of monomeric metal complexes. As a result of this approach, the formation of metal nanoparticles and the carbonized polymer matrix stabilizing them occurs simultaneously in one pot. It is shown that this approach makes it possible to obtain finely dispersed nanoparticles with a narrow distribution in size and volume of the matrix, as well as stable nanoparticles that can retain their structure and properties for a long time (at least after a year or more). In contrast to our previous works, in this work, the synthesis is optimized in terms of temperature (400 and 600 °C) and reaction time (2 and 4 h). In this work, the phase composition and microstructure of the obtained nanoparticles are also studied in more detail using not only XRD, but also TEM with SAED and EDX. Also, for the first time for these systems, their magnetic properties were studied. Relevant explanations have been added to the text of the article.

Why did you not perform the measurements above room temperature for the Fe-Co NPs?

Response: During ZFC-FC measuring, we have limited temperature range, since all the necessary features and critical points (bifurcation and blocking temperatures) for our samples, as well as for most magnetic nanoparticles, lie in this interval.

We have added the proper references and expanded the explanation part of the manuscript to guide the reader through the magnetism of synthesized nanocomposites.

Comments on the Quality of English Language

You have to reread the manuscript before the submission. There is a Russian text in lines 87 and 140-1 (what is the subject in the latter sentence?).  English should be checked. There are some parts with strange formulations and gramma mistakes: e.g., line 41 "ratio o " should be "of"; line 45 "able capable". Think also about the readability of your text, e.g., there should be some introductory sentence as "Figure 1 presents" in line 231.

Response: We apologize for the unfortunate errors made in the preparation of the article.

All your comments have been taken into account and corrected. In the text of the manuscript, all corrections are highlighted in color.

The conclusion is out of my understanding, it should be fully rewritten. However, it is understandable in case when the novelty and motivations are unclear. 

Response: Agree with your comments. Conclusion is rewritten.

The figures should be formatted the SAME way. You used different colors in similar Figs. 1(a,b) and 2(b,c). The latter is important because you used the second y-axis of the red color. The figure captions are not informatory enough, e.g., you forget to put the annealing temperature to Fig. 3(a). I recommend to add labels of the NPs for each figure as in Figs. 7-8.

Response: Agree with your comments. All figures are formatted according to your comments.

I also recommend to reorganize the text structure slightly. The Introduction is full of technical details you could move to the second material section. For the Material section, two subsections (synthesis and characterization) would be quite enough. Think to move subsections 3.1-3.3 also to the Material-synthesis or present separately. I lack this separation between the synthesis, experiments performed to check the synthesis and the results obtained for the synthesized NPs.  

Response: Thanks for your comments. Perhaps you are right. We will certainly take this into account in our future articles. But for the time being, we decided to keep the structure of the manuscript as a whole, but at the same time we removed the division into subsections. We left only two subsections (synthesis and characterization). We do not consider it appropriate to move Sections 3.1-3.3 to the Methods section. The latter is only an Experimental section where syntheses are given so that the reader can reproduce these experiments. Technical details have also been removed from the Introduction.

Reviewer 2 Report

The authors have submitted the manuscript entitled “Polymer-assisted synthesis, structure and magnetic properties of bimetallic FeCo- and FeNi/N-doped carbon nanocomposites” Some questions need to be answered.

1-      There are some English errors in the text, please check the manuscript (e.g., p1 line 41)

2-      The introduction section (p2, lines 51-55) “temperatures of 350, 400 and 600 °Ð¡” was repeated two times.

3-      There is a reference (lines 86-87) in the Russian language.

4-      In the Materials and Methods section (p2, lines 140-141) use the Russian language to describe reagents.

5-      The method used to calculate or found the composition for Table 1 is not mentioned.

6-      This method uses acrylamide to synthesize nanoparticles but is a very toxic material (Acrylamide was shown to be a neurotoxicant, reproductive toxicant, and carcinogen in animal species). It is a very dangerous route to synthesize material.

7-      The lattice parameter accuracy is very high concerning the XRD pattern. Also, the step time and step size of the XRD measurement were not mentioned.

8-      Both FeCo and FeNi have a hysteresis loop with a high value for Hc in the M-H curve diagram, so they are not superparamagnetic materials (for more information please check “Carbon-coated FeCo nanoparticles as sensitive magnetic-particle-imaging tracers with photothermal and magnetothermal properties”).

9-      The introduction section discussed about a very high saturation magnetization of FeCo but the result of magnetic properties was very low.

Author Response

Dear Reviewer,

We greatly appreciate your thoughtful comments that helped improve the manuscript.

Thank you very much for your effort.

In the following, we give a point-by-point reply to your comments:

Comments and Suggestions for Authors

  • In Materials and Methods and other parts of the article, the authors have used Russian words. please check it.

Response: We apologize for the unfortunate errors made in the preparation of the article. All your comments have been taken into account and corrected. In the text of the manuscript, all corrections are highlighted in color.

  • The language of the article in general needs a fundamental revision.

Response: Revision is done.

  • Add a schematic of the synthesis section to the article.

Response: It is done. A schematic is added.

  • In the X-ray diffraction pattern section, combine both analyzes in one figure.

Response: X-ray diffraction patterns of FeCo and FeNi nanoparticles are combined in one figure.

  • Is your synthesis an optimized synthesis? Are Fe and Co values optimized?

Response: Yes, the synthesis conditions are optimized in terms of temperature and reaction time. Relevant comments are included in the text. As for the ratio of the components of iron and cobalt (nickel), in this work only the ratio of 2:1 is taken. In subsequent studies, other ratios of these components will also be considered.

  • State the novelty aspect of your research in comparison with similar researches in the introduction.

Response: As mentioned in the text of the article, we have developed an original method for obtaining mono- and bimetallic nanoparticles by combining the polymerization process and controlled thermolysis of monomeric metal complexes. As a result of this approach, the formation of metal nanoparticles and the carbonized polymer matrix stabilizing them occurs simultaneously in one pot. It is shown that this approach makes it possible to obtain finely dispersed nanoparticles with a narrow distribution in size and volume of the matrix, as well as stable nanoparticles that can retain their structure and properties for a long time (at least after a year or more). In contrast to our previous works, in this work, the synthesis is optimized in terms of temperature (400 and 600 °C) and reaction time (2 and 4 h). In this work, the phase composition and microstructure of the obtained nanoparticles are also studied in more detail using not only XRD, but also TEM with SAED and EDX. Also, for the first time for these systems, their magnetic properties were studied. Relevant explanations have been added to the text of the article.  

  • Discuss in detail about the synergistic effect of FeCo/C-N components.

Response: Thanks for the interesting question. It's hard to say for sure. It can be assumed that such an effect is best manifested in the functional properties of such systems. In particular, in our previous work [Uflyand, I.E. et al. Preparation of FeCo/C-N and FeNi/C-N Nanocomposites from Acrylamide Co-Crystallizates and Their Use as Lubricant Additives. Micromachines 2022, 13, 1984, doi:10.3390/mi13111984] shows that bimetallic nanocomposites are more effective additives in improving the tribological properties of polyethylene composites than their monometallic counterparts. We expect the same effect in the catalytic properties of FeCo and FeNi nanocomposites; we are conducting such studies. As for the magnetic properties, for example, the Fe/C-N monometallic system does not exhibit superparamagnetic properties and behaves like a ferromagnetic material in a wide temperature range, interparticle bipolar interactions dominate. 

Reviewer 3 Report

An original approach to the synthesis of magnetic metal-polymer composites is proposed. Samples of two composites with a polymer matrix and FeCo- and FeNi nanogranules were prepared and characterized. In view of the growing interest in flexible magnetic composites, the goals and results of the work seem relevant and suitable for publication. The results are obtained by a worthy set of methods, clearly stated. The work can be accepted with taking in to account the comments below.

1. In fig. 5 and 6, the composite structure of the samples is clearly visible. Microelement analysis is also mentioned. It would be useful to show a composition distribution map or a line scan capturing the granules and matrix so that it is clear that the dark particles in the contrast images are indeed a metal alloy.

2. TEM images show that the volume fraction of granules in the FeCo-C sample is lower than in the FeNi-C sample. By the value of Ms, this fraction can be quantified as Ms/Ms_pure alloy (see, for example, [M.N. Volochaev… Phys. Solid State. 60 (2018) 1425–1431. https://doi.org/10.1134/S1063783418070302.]). This evaluation will show that while the concentration of metal granules in the FeCo-C sample is significantly below the percolation threshold, for the FeNi-C sample this concentration can be close to such a threshold. Did the authors check the electrical properties of the samples? I would suggest adding a short discussion of the volume fraction of granules in the samples. This may give additional confirmation to the authors' statement “... the obtained FeCo/N-C nanocomposites are superparamagnetic and exhibit exchange-bias behavior at low temperatures. In turn, FeNi/N-C nanocomposites are ferromagnetically ordered.”

3. In Table 2, there is a sloppy creeping of symbols. Check it.

4. In lines 86-87, 140-141, change the text to English.

5. p.376 “bimetallic metal nanoparticles” would be better replaced by “bimetallic nanoparticles” or “magnetic alloy nanoparticles”.

Author Response

Dear Reviewer,

We greatly appreciate your thoughtful comments that helped improve the manuscript. Thank you very much for your effort.

In the following, we give a point-by-point reply to your comments:

Comments and Suggestions for Authors

An original approach to the synthesis of magnetic metal-polymer composites is proposed. Samples of two composites with a polymer matrix and FeCo- and FeNi nanogranules were prepared and characterized. In view of the growing interest in flexible magnetic composites, the goals and results of the work seem relevant and suitable for publication. The results are obtained by a worthy set of methods, clearly stated. The work can be accepted with taking in to account the comments below.

 1.In fig. 5 and 6, the composite structure of the samples is clearly visible. Microelement analysis is also mentioned. It would be useful to show a composition distribution map or a line scan capturing the granules and matrix so that it is clear that the dark particles in the contrast images are indeed a metal alloy.

Response: Yes, you are right. We have confirmed the composite nature of the obtained materials by TEM, EDX, and elemental analysis data. Previously, we obtained a composition distribution map for FeCo/C-N nanoparticles [Dzhardimalieva, G. et al. FeCo@N-Doped Nanoparticles Encapsulated in Polyacrylamide-Derived Carbon Nanocages as a Functional Filler for Polyethylene System. ChemistrySelect 2021, 6, 8546–8559, doi:10.1002/slct.202101624.], and it was shown that the dark particles in the contrast images are indeed a metal alloy. Unfortunately, for FeNi/C-N, such studies are currently difficult, but such an analysis will be carried out in the future.

  1. TEM images show that the volume fraction of granules in the FeCo-C sample is lower than in the FeNi-C sample. By the value of Ms, this fraction can be quantified as Ms/Ms_pure alloy (see, for example, [M.N. Volochaev… Phys. Solid State. 60 (2018) 1425–1431. https://doi.org/10.1134/S1063783418070302.]). This evaluation will show that while the concentration of metal granules in the FeCo-C sample is significantly below the percolation threshold, for the FeNi-C sample this concentration can be close to such a threshold. Did the authors check the electrical properties of the samples? I would suggest adding a short discussion of the volume fraction of granules in the samples. This may give additional confirmation to the authors' statement “... the obtained FeCo/N-C nanocomposites are superparamagnetic and exhibit exchange-bias behavior at low temperatures. In turn, FeNi/N-C nanocomposites are ferromagnetically ordered.”

Response: Thanks for the interesting comments. We have not analyzed the electrical properties of such systems at this stage, but we plan to carry out such studies in the future. The work presented by you is interesting and useful, we have included in the text the relevant comments.

  1. In Table 2, there is a sloppy creeping of symbols. Check it.

Response: Done.

  1. In lines 86-87, 140-141, change the text to English.

Response: We apologize for the unfortunate errors made in the preparation of the article.All your comments have been taken into account and corrected. In the text of the manuscript, all corrections are highlighted in color.

  1. p.376 “bimetallic metal nanoparticles” would be better replaced by “bimetallic nanoparticles” or “magnetic alloy nanoparticles”.

Response: Done.

Reviewer 4 Report

1- In Materials and Methods and other parts of the article, the authors have used Russian words. please check it.
2- The language of the article in general needs a fundamental revision.
3- Add a schematic of the synthesis section to the article.
4- In the X-ray diffraction pattern section, combine both analyzes in one figure.
5- Is your synthesis an optimized synthesis? Are Fe and Co values optimized?
6- State the novelty aspect of your research in comparison with similar researches in the introduction.
7- Discuss in detail about the synergistic effect of FeCo/C-N components.

Moderate editing of English language required

Author Response

(The authors gave the same response as above.)

Round 2

Reviewer 2 Report

The authors have submitted the Revised manuscript entitled “Polymer-assisted synthesis, structure and magnetic properties of bimetallic FeCo- and FeNi/N-doped carbon nanocomposites” Some questions need to be answered. But they did not answered yet

1-       The method used to calculate or found the composition for Table 1 is not mentioned.

2-      This method uses acrylamide to synthesize nanoparticles but is a very toxic material (Acrylamide was shown to be a neurotoxicant, reproductive toxicant, and carcinogen in animal species). It is a very dangerous route to synthesize material.

2-      The lattice parameter accuracy is very high concerning the XRD pattern. Also, the step time and step size of the XRD measurement were not mentioned.

3-      Both FeCo and FeNi have a hysteresis loop with a high value for Hc in the M-H curve diagram, so they are not superparamagnetic materials.

4-      The introduction section discussed about a very high saturation magnetization of FeCo but the result of magnetic properties was very low.

Author Response

Comments and Suggestions for Authors

The authors have submitted the Revised manuscript entitled “Polymer-assisted synthesis, structure and magnetic properties of bimetallic FeCo- and FeNi/N-doped carbon nanocomposites” Some questions need to be answered. But they did not answered yet

  • The method used to calculate or found the composition for Table 1 is not mentioned.

Response: The calculation of the elemental chemical composition of the compounds was carried out according to their gross formulas for the compositions Fe3Co2C36H75N18O41 and Fe3Ni2C36H75N18O41; the experimental elemental composition of the compounds for C, H, and N was determined by organic microanalysis on a Vario EL cube elemental analyzer by burning the analyzed sample in the presence of an oxidizer in an inert gas flow. The content of Fe, Co, and Ni was determined by atomic absorption spectrometry on an AAS-3 instrument. During sample preparation, a weighed portion of the exact mass was dissolved using a mixture of three concentrated acids: HCl, HClO4, and HNO3.

  • This method uses acrylamide to synthesize nanoparticles but is a very toxic material (Acrylamide was shown to be a neurotoxicant, reproductive toxicant, and carcinogen in animal species). It is a very dangerous route to synthesize material.

Response: Yes, you are right about the toxic properties of acrylamide. However, it is well known that its polymeric analogue, polyacrylamide, does not exhibit the toxic properties inherent in its monomer. Acrylamide-based polymer hydrogels are widely used, including in biomedical applications [H. Bodugoz-Senturk, C.E. Macias, J.H. Kung, O.K. Muratoglu, Poly(vinyl alcohol)acrylamide hydrogels as load-bearing cartilage substitute, Biomaterials 30 (4) (2009) 589–596. https://doi.org/10.1016/j.biomaterials.2008.10.010; H.Ding et al. Polymer 171 (2019) 201–210. https://doi.org/10.1016/j.polymer.2019.03.061.; Si Yu Zheng et al. Macromolecules 2016, 49, 9637−9646. DOI: 10.1021/acs.macromol.6b02150. and others]. In our case, polymer-mediated synthesis involves the formation of a metal-polyacrylamide polymer product as an intermediate from which the final nanocomposite material is formed.

  • The lattice parameter accuracy is very high concerning the XRD pattern. Also, the step time and step size of the XRD measurement were not mentioned.

Response: Spectra for samples with Fe-Ni and Fe-Co particles were recorded at a constant rate of 1 deg/min, the step was 0.005 degrees. In our case, the error of the lattice parameter was determined by the calculation method. The experimental and model spectra were fitted. From the model spectrum in determining the center of gravity of the 2θ angles, the statistical error turned out to be low, hence the low error in determining the lattice parameters.

  • Both FeCo and FeNi have a hysteresis loop with a high value for Hc in the M-H curve diagram, so they are not superparamagnetic materials.

Response: We completely agree with the referee, and we have improved the text of the article concerning magnetic properties during the previous round. However, it was our mistake that we did not exclude the superparamagnetic characteristics of the obtained nanocomposites from the Abstract.

  • The introduction section discussed about a very high saturation magnetization of FeCo but the result of magnetic properties was very low.

Response: The FeCo alloy in general possesses the highest possible magnetization at room temperature. However, in this study, we explore the N-doped carbon-metall composite, and the presence of the metallic nanoparticle is less than 50%. That is why the magnetization of our composites is very low. In addition, due to surphase effects, the magnetization of nanoparticles is lower in comparison with bulk samples.

Reviewer 4 Report

Accept 

Minor editing 

Author Response

Dear Referee,

We greatly appreciate your valuable comments that helped improve the manuscript.

Thank you very much for your effort.

In the following, we give a point-by-point reply to your comments:

Comments and Suggestions for Authors

Accept 

Comments on the Quality of English Language

Minor editing 

Response: Done